# You can't report your feelings: The hidden labor of managing threats to safety by women in global public health fieldwork

**Corey McAuliffe**[1]*, **Ross Upshur**[1], **Daniel Sellen**[2], **Erica Di Ruggiero**[1‡]

**1** Dalla Lana School of Public Health, University of Toronto, Toronto, ON, Canada, **2** Joannah and Brian Lawson Centre for Child Nutrition, University of Toronto, Toronto, ON, Canada

‡ EDR is the senior author
* corey.mcauliffe@ubc.ca

**Data Availability Statement:** The data that support the findings of this study are not publicly available due to the ethics consent form indicating that

## Abstract

Increasing job market demand for and availability of Canadian and U.S. global academic health programs in post-secondary education increases student demand to participate in internationally based fieldwork, while supportive resources remain weakly developed. Previous studies indicate provisions to protect the health, safety, and well-being of women students remain inadequately addressed during training, while more research to identify needs, expectations, gaps, and best practices would inform policy and practice to improve conditions for women working off-campus on global public health studies. One approach, reported here, is to document and better understand the lived experience of U.S. or Canadian women graduate students participating in global public health fieldwork. Participant in-depth phenomenological interviews and guided writing exercises aimed to capture lived experience descriptions for 25 women. A phenomenology of practice was applied throughout the research process, following Max van Manen's qualitative methodology approach. Loss of environmental familiarity, combined with graduate students' lack of power, created considerable hidden labor described by women in working to keep themselves safe from sexual and gender-based violence (SGBV) while participating in global public health fieldwork. Women shared specific experiences exemplifying how this can be both alleviated and/or intensified through a range of negotiated strategies, coping styles, and management techniques. Additionally, women recalled laboring as students to avoid or reduce instances of SGBV, that then, precluded them from having any material "of substance" to report once returned home. These findings offer new meaning structures, language for a foreign experience, or ways to describe, conceive of, and respond to global public health fieldwork that hold the potential to positively affect individuals' experiences, institutional understanding, and thus practice, of future women students in global public health.

participant data would not be shared outside of the research team. These data may contain information that could compromise the privacy of research participants, and thus will not be made publicly available. This research study was approved by the Research Ethics Board at University of Toronto. If you have any questions or concerns or requests for data access, please contact the Research Oversight and Compliance Office – Human Research Ethics Program at ethics. review@utoronto.ca or by phone at +1 (416) 946-3273.

**Funding:** The authors received no specific funding for this work.

**Competing interests:** The authors declare that they have no known competing financial interests or personal relationships that could have appeared to influence the work reported in this paper.

## Introduction

*Trigger content warning*: this article includes content and references pertaining to traumatic and distressing experiences of individuals and specifically focuses on experiences of sexual and gender-based violence.

The field of global health has grown steadily since the turn of the century. Global health programs, departments, centers, and certifications in higher education have continued to multiply across Canadian and U.S. academic institutions. This increase has translated into higher student demand to participate in global based fieldwork for practicums, research, and dissertation projects [1] within a Global South that has been historically oppressed and colonized. Schools of public health are one of the prominent spaces where students participate in global health fieldwork, sending thousands of students, a majority of whom self-identify as women, to the Global South yearly [2,3]. Limited research suggests a wide range of both negative and positive short- and long-term implications which necessitate further exploration [4,5]. Moreover, insufficient research has been dedicated to understanding the physical, mental, and emotional lived experience (and thus, consequences) of public health graduate students participating in global health fieldwork, including women's experience of sexual and gender-based violence (SGBV) [3,4,6,7].

### Women's experience of sexual and gender-based violence

A 2018 Workshop on Ethically Managing Global Health Fieldwork Risks, co-sponsored by Agnes Scott College, The Taskforce for Global Health, and Emory University Rollins School of Public Health, identified gender-based violence during fieldwork, alongside a culture that then compels silence, as the most concerning theme. Within the academic and global health implementation-oriented group, almost every woman present had been affected by gender-based violence, while many men had not realized the severity or extent of the issue [3]. Many have called for the field of global health to more intentionally focus on the protection of women in academia from experiencing SGBV, both locally and globally [3,6,7]. Women across disciplines describe experiences of unwanted physical contact, sexual harassment, intimidation, and sexual assault by supervisors, colleagues, gatekeepers, participants, acquaintances, and strangers [4,5,8]. According to Clancy et al. [8], sexual harassment and assault during fieldwork was most often aimed at students and postdocs, with existing literature demonstrating detrimental and long-lasting impacts on future careers [9]. Furthermore, we previously reported that women's experience of SGBV can lead to feelings of "fear, depression, anxiety, isolation, self-blame, and PTSD" [10], identifying many harmful long-term effects. Graduate students experiencing research-related trauma constitute "a high-risk group" due to additional strains such as completion deadlines, inability to take breaks from fieldwork, and funding constraints [11]. Consequently, students may further worry about future career prospects and their reputation [12].

### Hidden labor

Labor, a form of hard work, includes intentional physical or mental activity with the purpose of economic or emblematic gain and is inclusive of many forms of hidden labor (e.g., care work, self-employment, etc.) [13]. More recently, disability scholarship critically explored hidden labor, focusing on its disproportionate distribution [14,15], as it requires "added effort, time, and consideration" and is "conceptualized as 'hidden' insofar as [it is] not formally recognized as 'supplementary work,'" but rather seen as innate responsibilities [16]. It is inclusive of a "disproportionate investment of physical and emotional energy" due to "strategies such as

concealing information to avoid stigmatization or suppressing and counterfeiting particular emotions" that "also have a psychological impact on a person's sense of self" [13].

Within their formal program requirements, graduate students undertake a great deal of mental, emotional, and physical labor, exacerbated by time-constraints and financial costs [17,18]. Moreover, students' labor can be hard to define and is highly precarious, with few to no policies guaranteeing legal protections [6,10]. Hidden work is not equally valued and makes advocating for better working conditions or protections extremely difficult [19]. Students' work can incorporate a mix of paid and unpaid labor, muddled by expectations attached to fellowships, scholarships, and stipends. Often framed as an internship, trainee role, or volunteer position, students' work is purported to offer skill acquisition and "the moral prestige of the organization and its mission" [20]. This conceptualization allows institutions to reap far more benefit and reward than students when labor is free/underpaid.

The impact of hidden labor, specifically on women in global public health fieldwork, has not been an explicit focus of previous public health research. Authors, often speaking about their own graduate student experiences, have implicitly shared hidden labor endured, through stories of SGBV during global health fieldwork and the aftereffects, e.g. [21–24]. Dealing with and avoiding SGBV demands considerable effort and requires a substantial amount of emotional and cognitive resources [15]. Moreover, women disproportionately face SGBV within global public health fieldwork, especially as students [8]. The negative consequences associated with SGBV, and the undertaken labor further disadvantages them when reacting to and trying to avoid further SGBV experiences. However, this has not been widely assessed, especially within the global public health field, and requires further exploration. Thus, our central aim was to explore how U.S. and Canadian women graduate students experience global public health fieldwork. We sought to understand the opportunities, difficulties, and challenges of women's safety in fieldwork, while aiming to create open and authentic discussions.

## Methodological approach

Our study is situated within van Manen's [25] qualitative methodology, a phenomenology of practice. Merleau-Ponty [26] and van Manen's [25] guided existential reflection (relationality, corporeality, spatiality, temporality, and materiality) informed the research process. Our interpretive approach aimed to better understand the lived experience of women graduate students participating in global public health, privileging participant knowledge through their experience of living- or being-in-the-world. Thus, "the focus of a hermeneutic inquiry is on what humans experience rather than what they consciously know" [27]. Through the illumination of lived experiences, our intent was to find what is implicit and obscure.

### Methods

**Study setting and participants.**  Data creation was conducted with participants in the U.S. and Canada between January and October 2018. Inclusion criteria included: self-identification as a woman; being from the U.S. or Canada; studied at the graduate level (Master's or doctoral) at a U.S. or Canadian academic institution; conducted graduate-level global public health fieldwork for at least 4 consecutive weeks, in the Global South, between 2000 and 2016. There were no exclusion criteria.

We recruited a purposive sample of participants through email and social media, based on the authors' global health networks. This included reaching out to various Canadian and U.S. schools of public health (e.g., deans, faculty, alumni) and established associations, including student groups connected with these organizations (e.g., Consortium of Universities for Global Health, the now former Canadian Coalition for Global Health Research). Forty-nine women

agreed to participate; four were ineligible. Those eligible were invited to participate in multiple in-depth phenomenological interviews (IDPI) or in a guided writing exercise (GWE). Seven women were selected and participated in two to three IDPIs, ranging from 90 to 210 minutes each. Eighteen women elected to participate in the GWE. In keeping with phenomenological methodology, sociodemographic information (e.g., race, age, sexual orientation) were not collected. More than 20 different countries were identified within the 25 participants' stories, including countries in sub-Saharan Africa, Latin America, Middle East, South Asia, and Southeast Asia. Participants attended 14 different universities, completing their global public health fieldwork between 2010 and 2016 at a master or doctoral level.

**Data creation, analysis, and positionality.** For the IDPIs and GWEs, participants were given a brief background, rationale for the phenomenon of interest, and the same overarching research question: *What was it like to practice global health abroad during your graduate public health training*? IDPI participants were then given time to reflect on their experience and asked to create a timeline or identify key points to reflect upon as an elicitation tool. Overarching questions were derived from what was brought up in the participant's timeline or by what may have been missing (based on experiences brought up by other participants). Follow-up interviews took place after data had been transcribed and reviewed by the primary author. The ability to follow-up with participants allowed for the opportunity to build trust between the participant and the interviewer, to remember experiences over a longer period, and to use various elicitation and reflection methods to access knowledge, memory, and meaning in a variety of ways. GWEs captured lived experience descriptions in a written format where participants were asked to think back to a moment when they were a graduate student participating in global public health practice and to write a concrete account of what it was like. They were further encouraged to try not to explain or interpret their experience; rather, to only describe the moment in as much experiential detail as they could recall. Participants could submit as many stories as they would like.

The creation and analysis of phenomenological data are an inseparable part of a holistic and iterative process [28]. Participants shared their lived experience and reflective understandings, while the primary researcher sought to comprehend, rather than challenge, their perceptions [29]. Data creation involved detailed descriptions of lived experience, rather than cognitive reflections on what participants thought about the experience [25]. The goal was to discover commonalities and differences in "culturally grounded meanings," rather than understand individualities or idiosyncratic events [30].

IDPIs and GWEs were intentionally used "as means for exploring and gathering experiential narrative material. . .for developing a richer and deeper understanding of a human phenomenon" [28]. The findings below are crafted stories that have been (re)storied and aligned to create *a woman graduate student's story*. This story is based on individual- and composite-based stories from 25 women. The story, as recounted in the findings, takes many forms, shifting relationally, temporally, materially, spatially, and corporeally. The story incorporates a woman graduate student's emotional, mental, and physical concerns regarding SGBV during her fieldwork. She is at once any global public health woman graduate student and yet, no *one* graduate student, to treat "all understandings as culturally and historically situated, exploring multiple perspectives and possible meanings" [31]. At the core, the story speaks to the potential experience that we each have had, could have, or could see ourselves having if standing in her shoes [28]. Our use of exemplars thus aims to "characterize specific common themes or meanings across informants" [32].

The story below incorporates the experiences of women from various racial, country, and cultural backgrounds, including lesbian and queer women. While participants were not asked to explicitly identify their ethnic or racial identity or sexual orientation, at times, their identity

emerges through particular experiences. Based on identities elicited from the data, it appears that most participants are able-bodied, cis-gender, heteronormative, or White/White-passing women. As the co-creator of this phenomenological story, the primary researcher's positionality as a White, cis-gender, heteronormative woman with an episodic disability inherently impacts the way these stories are shared. While derived from participants, the method of (re) storying inherently weaves the lead author's voice throughout the narrative.

**Ethical considerations.**   The University of Toronto's Research Ethics Board approved this study. Formal informed written consent was obtained at the beginning of the study, and the process of consent was understood as ongoing and revisited at each interview. Participants were free to withdraw at any time; however, no participants chose to withdraw.

## Findings: The hidden labor of safety for a woman graduate student

### Overview of findings

A woman graduate student, henceforth referred to as "a student," consistently experiences SGBV while participating in global public health fieldwork. Her loss of environmental familiarity and her subordinate role as a student, creates a significant burden of hidden labor that can be either alleviated or intensified due to negotiated strategies and management techniques. In the end, her labor to avoid or reduce instances of SGBV may preclude her from having anything "of substance" to report. Based on the consistency across study participants, her story is one that most women in global public health likely relate to.

When one hears of a student participating in global health fieldwork, the most acknowledged stories are those that incorporate positive, life-altering experiences–where a student takes difficult and challenging situations and demonstrates how the experience serves to foster valuable insights for her future career. These stories are the ones featured in fellowship and job applications, institutional news bulletins, and marketing materials. They are the stories that a student is trained and learns to tell.

While these stories can be authentic, the story often silences or suppresses how a student experiences and embodies safety-related challenges and traumas. These experiences lead to a myriad of concerns for a student; yet her whole story is incompletely acknowledged. While general safety concerns affect all students participating in global public health fieldwork, there is additional complexity to the sheer volume of labor expended and required for a woman student to keep herself safe from SGBV.

While the phenomenon of hidden labor in response to SGBV is not exclusive to women graduate students, their experience is distinct through two intersecting influences (Table 1)– loss of environmental familiarity (i.e., preparation, travel safety, language skills, etc.) and that students are in inherently subordinate positions (as compared to faculty, administrators, etc.). This can leave a student feeling like she has a lack of power, due to her precarious work status, compounded by her gender, age, work experience, financial precarity, research funding status, and/or previous SGBV experiences. With her labor frequently and habitually hidden, she feels her burden is highly inequitable as compared to male peers doing similar fieldwork.

A substantial burden of hidden labor occurs because of these two influences, as a student works to adapt and negotiate new strategies employed to keep herself safe (Table 2). She performs a considerable amount of hidden labor in the form of "*managing*" people and situations she finds unsafe, as well as managing her own mindset through coping and reframing her experience.

The implicit nature of considerable SGBV makes her labor difficult to identify, as much of this work is normalized and internalized. Much hidden labor is invisible to those who do not experience daily SGBV. However, the same invisibility may exist for the student herself, as she

**Table 1. Creation of hidden labor in global public health fieldwork.**

| Two Influences Converge: | |
|---|---|
| **Loss of Environmental Familiarity** | **Subordinate Position as a Student** |
| • *Preparation and follow-up*: insufficient, not specific to gender or intersectional identities<br>• *Travel safety*: traveling/walking solo, public transit, unsafe transport conditions<br>• *Loss of peer support*: difficulty making local friends; isolation; reduced connection to those at home<br>• *Language*: miscommunications, inability to communicate effectively<br>• *Cultural influences*: intersectional identities compounds hidden labor; added and protective measures and challenges (e.g., living between two worlds, differing expectations) for those with in-country familial origin | *In-country work environment*<br>• SGBV perpetuated by those in power (e.g., colleagues, in-country support, participants, others tangential to work)<br>• Lack of supervision/mentorship<br>• Student responsibility to keep herself safe<br>• SGBV targeted at local women<br>*Financial insecurity*<br>• Lack of funding (e.g., debt due to airfare, living expenses, tuition)<br>• Authentic experience–need to obtain cheapest accommodations and travel<br>• University safety policies cause increased barriers and costs<br>• Underpaid or unpaid practicum/work<br>• Increased labor once home: pay off debt |

may have grown up or been trained in a way that normalizes these experiences. Furthermore, her subordinate position as a student and less familiar environment are at the root of why she infrequently reports SGBV both during and after her global health fieldwork (Table 3). This specific experience of global public health fieldwork does not negate a student's experience of SGBV at her home university. Her experience back home, both pre- and post-fieldwork, also produces hidden labor that she must learn to navigate. Instead, this story is an opportunity to explicitly acknowledge additional work performed during fieldwork to keep herself safe.

## Loss of environmental familiarity and subordinate position as a student

Much of her hidden labor is due to converging influences of losing one's environmental familiarity, alongside her subordinate position as a student (for a more thorough assessment of these experiences see [33]). Overall, a student understands she will have less familiarity with language, cultural norms, and transportation systems, as compared to her home. However, there is a broad continuum for the extent to which she feels a sense of unfamiliarity and newness in her fieldwork site. A student may return to a country where she was born, a country of ancestral lineage, somewhere she visits frequently, or somewhere she has never been. Her ability to communicate in the local language varies, as does the ability for others to speak to her in English. She attempts to dress and act in culturally appropriate ways, many of which are different from back home. These considerations, in turn, impact the level of support and comfort she feels while doing her fieldwork.

**Table 2. Influences that create increased and/or decreased hidden labor for women.**

| Negotiations and Strategies |
|---|
| • Support network: University, work, and strangers<br>• Support network: Partners, peers, and roommates<br>• Language, clothing, and appearance<br>• Managing and laying traps: Avoidance, isolation, vigilance<br>• Mindset: Questioning, coping, reframing, setting boundaries |

**Table 3. Influential structure on women's decision of whether to (not) report SGBV.**

| (Not) Reporting SGBV |
|---|
| • Close peer and classmate debriefs |
| • Supervisor: Helpful and harmful responses; expectation for student to bring up concerns |
| • Transition from Student to Worker: Power dynamics shift; more able to speak up/out |
| • Students in Similar Placement: Inability to share if SGBV has not been reported |
| • Can't Report Feelings: If no specific "incident", inability to share experience |

No matter the student's background, she experiences a shift in her accessibility to resources and ease with which she interacts with her support network during fieldwork. These support networks are made up of her home university, local in-country institution, local community members, peers, and family members. A student's story foundationally rests upon some loss and shifts to her support network, as she attempts to keep herself safe and well. Furthermore, feeling a lack of power or powerless in her workplace, creates additional labor in protecting herself. She struggles with a sense of control manifested through SGBV at work, little oversight by her in-country supervisor, financial insecurity, and a sense of personal responsibility to keep herself safe.

## Hidden labor through negotiations and strategies

While a woman student's unique experience of doing fieldwork can lead to experiences of SGBV, she is not just a passive experiencer of SGBV. Starting in childhood, structural and environmental gendered norms shape her experience. Once she recognizes inequities within her work environment, she begins to trust her gut and intuition, homing in on helpful negotiations and strategies to reduce/prevent SGBV. These strategies include cultivating environmental familiarity (see Table 1). As her mindset shifts to avoiding and "managing" people and places she interacts with, she further strategizes how to cope and survive. Some choices are explicit, made in advance or momentarily, while other decisions are circumstantial, implicit, or instinctual. All decisions are affected by the resources and support networks available to her, which can be protective (mitigating additional labor) or harmful (creating additional labor). Moreover, an increase and decrease of labor can occur simultaneously, such as having a male colleague present during a precarious situation (protective), while she is concurrently mindful of her actions to not be misinterpreted as sexual overtures (negative). These negotiations and strategies point towards the compounding and multiplicative effects of hidden labor performed and towards opportunities where a student needs better support to allow for a sense of safety and security. Having to manage and control how she moves her body within fieldwork uses an immense amount of labor, including time, finances, and mental awareness.

**Support network: University and work.** Support networks are the most common relational strategy to keep any student safe during fieldwork. When a student has a good support system in place, this network offers cautions and safety advice such as traveling with others, calling someone to pick her up, paying extra money for more direct transit, and not traveling after dark. However, this advice still requires labor, as she is again responsible for keeping herself safe.

A student's supervisor plays a large role in the creation of appropriate and supportive networks before, during, and after her fieldwork experience. While her supervisor may (or may not) do a thorough job of connecting her to appropriate and opportunistic projects, a student feels supported when her supervisor accompanies her on visits. The supervisor's presence

assists in connecting her to an in-country support system, offering an additional layer of security.

*My supervisor who is very engaged is there for the first week we are in the field, getting us set up. This is absolutely critical, because we are in this totally new context, very far from everything else. She has lived and worked there for years. She knows everyone and is able to introduce us.*

A student can further feel supported and protected by her work colleagues or team members, such as a male research assistant, who may provide added protection and safety, especially in countries where she cannot be outside without a man's supervision. However, while feeling supported by work colleagues or team members is not always her experience, she recognizes the importance and level of support this could offer. When forced to travel alone, her experience is varied. She is grateful when strangers come to her aid but having to rely on others produces anxiety, and helpfulness is not always her experience.

*The country is infamous for being a dangerous place for women. The endless warnings to be careful and the constant fear and need for vigilance take their toll. While working to be careful, I feel very lucky to travel with teams almost all the time. One time I have to return alone by bus late at night. A drunk man sits beside me, moving his thigh against mine and bumping his hand onto my lap as he reaches repeatedly to adjust the sliding window. Eventually another man orders him to switch seats. The new man sits beside me and leaves me alone the entire ride, only checking to ensure I have a safe ride home at the end.*

Beyond a sense of physical protection, a student also relies on her team members and colleagues for emotional and mental support. Debriefing with them gives her space to feel supported and heard, building a sense of team morale and togetherness. When she faces personal challenges or isolation, being able to rely on colleagues allows her to feel cared for and protected. Unfortunately, this is not always her experience.

**Support Network: Partners, peers, and roommates.** At times, a woman student identifies having a male partner, ranging from fictitious (to dissuade aggressive flirtations) to having her partner accompany her during fieldwork, offering another layer of protection. The creation of a fake partner further protects lesbian and queer women in homophobic spaces. If further physical protection is necessary, she recruits men to pretend to be her partner. Partners who remain back home, allow for a daily check-in. Personal questions asked of the student extend beyond questioning her relationship status, also focusing on whether she has children, and if not, why, and when she will. The constant expectation for her to have a particular type of family is exhausting.

*"Why aren't you married? Why don't you have kids? When are you getting married?" The cultural differences are huge. Having my husband there helps me to build different relationships and prevents misinterpretation of intentions, I don't have to worry about how I am perceived when being friendly or nice. But before my husband arrives, there are suggestive comments from my research coordinator and really flirtatious men, which doesn't happen once my husband is there. Even if I say I am married and clearly draw those boundaries, people continue to flirt.*

A student feels safer and more protected when making local and expatriate friends, including people she knows from back home. Within this peer group, locals (e.g., family, friends,

roommates, colleagues) offer a level of safety and protection through their environmental knowledge and awareness. Furthermore, peers allow her to go out accompanied, offering space to share and debrief about experiences and feelings.

*I have it good. I make friends with people at the NGO, make friends who are from there, as well as with the expat community. By the end, it is easier to go out and do stuff. But at first, it is suffocating. But what's my alternative? Sit in my room and be totally lonely*?

Having a roommate can also add to her safety, especially when they act as a cultural interpreter and offer connections to local friend or family groups.

*I have a great roommate who is also a student from my university. She is from the country we are in and is basically my cultural translator. Any time I go out with her, she is helpful. When we are on a rickshaw, we stop at this traffic circle and wait 10 minutes to move. Inevitably, people come and beg, sometimes aggressively. Because she can communicate with them, they say, "She's from here," and we're suddenly off-limits. She is this buffer between me and what I perceive to be a risky situation. I become super close to her and her family. I learn a ton from them. They make me feel at home. It makes it less lonely.*

Friends and roommates provide safety while traveling, as noted above. This safety extends to the idea that it is safer to travel in numbers. From the beginning of fieldwork, a support network is helpful, as someone trusted can pick her up when she first arrives, lessening stress and anxiety during her transition. While in country, friends provide her with someone to call, in case of emergency. Her peer group also offers someone to debrief with, important for her well-being. These friendships offer protection from potentially dangerous situations during data collection.

*There is another student who lives nearby. We go for walks, debrief, and escape at times. It makes me feel safer to be with her. She and I spend the whole hour and a half talking, decompressing, and sharing feelings about challenges we are having. Sometimes we do our interviews together, waiting for the other to finish. I feel better when it is the two of us.*

**Language, clothing, and appearance.** A woman student further strategizes ways to keep herself safe through personal techniques, such as language skills, dress, appearance, and intentional spending. While her inability to speak the local language is associated with a less familiar environment, her ability to communicate in the local language helps keep her safe. It allows her to create stronger social support networks and integrate within communities. Furthermore, it enables her to communicate with locals, who help protect her from danger. The ability to communicate and having long standing ties to community members can also allow access to safer and more affordable accommodations.

*It is easy for me to integrate into this completely random and foreign culture because I speak the language. It is interesting to talk to the locals and to be able to quickly and easily integrate with very warm and open peopleE. I talk to all the shopkeepers, everyone at the restaurants, anytime I hop in a cab. By befriending different restaurateurs, if I notice someone following me, I walk right past my house and go hang out at the bar.*

Even if she is not fully fluent or proficient, her attempts to speak the language open spaces for social connection at work, especially with humor and light-heartedness. The ability to

communicate comes in handy when she is forced to diffuse sexual harassment, at times surprising people with her knowledge of the language and at other times feigning ignorance, to not engage. While ignoring and not engaging is often preferred, this conscious choice takes labor that often goes unrecognized. Moreover, daily labor goes into how she chooses to dress and appear to others. When she does not dress in the traditional style, she is policed by others telling her to better cover herself. She uses a variety of personal strategies; however, at the root, she tries to shape her body to help her blend in or become invisible.

> *I buy pepper spray and clothes that are too big for me, including maternity pants. I wear very loose clothes, as I think, "If I'm less visible, I may have less problems." Part of me wonders, "Is it worth it to make myself this small?"*

She further works to conform to cultural norms through body language and communication style.

> *I dress carefully and shape my body language to be shy and small like the women here. Since my language skills aren't great, I say far less than I normally would. I'm usually loud, bubbly, and a very expressive person, but I have become quieter and slower paced in the way I talk. I change my behavior. I feel like I'm asking women to share of themselves, but I'm not letting them really see me at all.*

**Managing and laying traps.**   The stories above elicit a myriad of ways in which the student uses her networks and ingenuity to prevent SGBV. She endures many forms of SGBV; yet she constantly works to *avoid* it, *manage* it, or *lay traps* to keep herself safe.

> *I lay traps to catch abnormal stuff. I take a weird path that's not a logical route, so if someone is following me for more than a block or two, I know they're doing it on purpose. I change my route to and from home, cutting through the back of a restaurant where their outhouse opens into an alley. I hop the fence to get to my home from there. I work to minimize my risk; I am as vigilant as I can be. I do things to catch the visceral stuff, I lay the crumbs. I ask a question and if their answer is a little off, I notice. Then, I extricate myself before I'm cornered or stuck in a situation I can't get out of. There is that visceral reaction, but also conscious planning to try and avoid situations where I will be in an extra vulnerable position.*

Sometimes the laying of traps or management is a conscious choice, while at other times an instinctual gut or intuitive response. Her experience is imbued with a need for hypervigilance and assuming the worst.

> *I can't tell you he did this specific thing to me, as I can with the past people. But the reason nothing happens is because I don't let it happen. I learn to listen to myself. I learn to trust myself. I learn to manage him, or I learn to manage my circumstances. If I'm walking a certain way and he's coming, I turn and walk the other way. My way of dealing with it is avoidance. I do that for six months. It's a lot of work to avoid people, but it's the price I pay to do the work I want.*

While working to avoid certain situations and people, she also does her best to not walk or travel alone at night. However, this direct impact on her behavior can cause her to isolate or miss social activities in her community or with colleagues and friends.

*At the beginning we are 15. I have someone walk me home if something feels sketchy or off. We go in packs, but after they all leave, it is just me. I either end up going home by myself in a cab or else I walk home if it is still light out. Mid-day I either go to a store or a restaurant to pick up two meals worth of food. My social interaction is limited to the time I spend sitting at my desk working and speaking a different language. Then, I go home and eat leftovers and freak out every time I hear a twig break outside. It feels urgent to not be out after dark.*

Unfortunately, there are times when a student has no other option but to traverse into an experience that may be dangerous or risky. She relies on her ingenuity and vigilance to mitigate harm at the expense of much labor. Learning from previous experiences, she often gets a gut instinct to not be alone with a specific person or to remove herself altogether. These strategies often conclude with her ignoring or avoiding someone or a situation altogether. The easiest course of action is often for a student to fully remove herself from a potentially harmful situation, which may be as basic as not going out after dark alone or as severe as damaging work and future career prospects to avoid specific meetings or social get-togethers when particular people are present.

**Mindset.** As the experience of SGBV is rarely discussed prior to departure, part of her hidden labor is interrelated with mindset. At times, she wonders if she's being naïve or potentially playing into a racist mindset, making her question her response to a situation in which she has little to no guidance. She also identifies how to cope and reframe her experience of SGBV. These may feel like necessary coping mechanisms to move forward but may ultimately be damaging or unhealthy.

*I start to feel like there's no interaction between a male and a female in this country that is professional. But I like global health. I don't want to burn any bridges. So, I don't say anything about it. It is how I process the situation and walk away from it. I see this experience as a steppingstone to what I want to do. I think, "This is what I have to put up with for two months for me to be able to have this experience." That's how I rationalize it in my mind. The harassment is what it is. It is something to get through for a bigger goal.*

*After my fieldwork, I am commonly asked why I don't just leave. It doesn't even cross my mind. I just want to be done with it, and I am merely trying to cope. I spent a year preparing for my study, setting up my community contacts, months of getting ethics approval, and then months of language lessons. I set up the entire study myself, and I am in the middle of it. It is impossible to step outside of the distress I am experiencing and leave.*

Her reframing incorporates experiences of precarious situations, where she states outright how she was frightened or felt in danger. Then, in the next breath, she expresses that the experience was fine. This reframe of *it's fine* is tied to what a respondent called a *silver lining mantra*, where she identified ways to cope to move forward.

As a student experiences situations that do not align to her own cultural values, she begins to shift her responses. While she works to be as culturally sensitive as possible, at some point, it becomes too much.

*There are constant moments where someone is ashamed for me. I'm covered, I'm sitting here doing my work when a man is present. I am constantly protesting something. I try to be respectful, but the things they try to get me to do don't make any sense to me. I want to be culturally sensitive, but I feel irritated, angry, and pissed off. I refuse to listen.*

These details become more apparent as she begins to recognize the deep inequities between her and her male university colleagues. Within a similar context, these students have very different experiences. She wonders why this is not explicitly talked about at her home university, and why this inequity is deemed acceptable.

*What makes it clear to me is that there is a male who does similar research to mine two years after I am there. We have many opportunities to debrief as he comes from the same university and supervisor. The overwhelming differences between our experiences are those barriers I face, as a result of being a woman, that he does not. Nobody cares what he wears, who he speaks to, or how he speaks, including how loud or the tone of voice he uses.*

As she questions her daily experiences, she becomes more and more uncomfortable with all the SGBV she encounters. She also may not question certain situations until years later. At this later stage, she looks back on her experiences with new knowledge and framing.

*I am shocked and surprised when he kisses me. I feel betrayed and confused because I don't see it coming. I feel nauseated thinking about it. It's not something I've talked about since. Now, with #MeToo, I think about it differently. At the time, I know that it happens, but I don't realize how common it is. Looking back, I end up doing the classic thing of blaming myself.*

After reaching out, via email, to a past colleague she had during her fieldwork placement, to inquire about potential jobs, a woman (now no longer a student) continues to face SGBV. With ample supportive work experience, she realizes how unacceptable the treatment she faced was during her fieldwork.

*The harassment from an email I receive makes the conversation more have-able. I now know, this is not just in my head. Anybody who sees this email can say this is wrong. All the other experiences are just me, questioning a person in power, with nothing to show for it. It has been 10 years, including four years of employment where I am respected, my ideas are valued. To go from that to the shit I'd left behind, I know it's not acceptable. The email makes me no longer want to work abroad. I realize, this is what I will have to go through constantly. I don't want that life.*

## (Not) Reporting

A too often silenced and suppressed story, SGBV is consistently perpetrated on a woman student. Embedded throughout her day, she has the unfortunate onus to negotiate and strategize her safety during fieldwork. The experience of workplace SGBV occurs much too frequently, and she is often at a loss for how to report these behaviors. Furthermore, she questions if anything can or would be done. She typically does not want to report her experiences to her university but is able to debrief with her peer group and classmates.

*There isn't an official debrief, but we do a lot of our own. I am still really close with some of the students I traveled with. We went through some intense experiences together. People have a fair amount of trauma they shared, and we give each other support. In a whisper we share tales like, "Remember that time? Remember this?" We have group threads over text message where we have in-jokes. We still group message (years later).*

At times, a student goes to her supervisor to discuss experiences of SGBV, where she is met with a range of responses–many of which, although not all, are inappropriate and harmful or

support that would have been helpful years prior. The onus is on the student to come forward with her experiences, even though she has little knowledge of how her supervisor will respond and whether it may harm future career aspirations.

> *When I talk to one of my supervisors upon returning, I tell her things have not gone well. She is surprised and tells me how she has spent time in another nearby country with her husband, and how things had gone really well for her. She says, it may be better if I have a man with me. This conversation deeply upsets me and makes me question if I need to be straight to work in global health.*

> *When I tell my supervisor about it, she also shares her unpleasant experiences as a woman of color. Her husband is white, and she is brown. She tells me, "Yeah. They used to think I was his prostitute. They all thought he paid me to be with him." But that was not discussed before I left, rather we discuss it four years after I come back. It doesn't come up, until I bring up my issues with her.*

As noted, a student lacks an equal amount of power, as compared to faculty, administrators, and supervisors. However, once a student transitions out of her student role these power dynamics shift. When able, she tells past supervisors and mentors more readily about her student experiences of SGBV.

> *Both of my co-supervisors are supportive women, but there is a power dynamic as a student. This makes it very difficult to share the extent of my trauma, so I downplay it. It feels too personal to share. Once I am no longer a student, I feel the power dynamic shift. Nothing changes on their end, but I feel I can be more outspoken. I work for both of them as a research assistant and feel more of an equal. Over time, I slowly share how it was a truly awful experience.*

While a student finds it hard to share her experiences with faculty, at times, she chooses not to share her experience of SGBV with future students embarking on similar placements. This leads her to feel guilty and question whether she is perpetuating the cycle, a form of self-blame and misplaced sense of responsibility.

> *Before leaving on my practicum, people from the fellowship meet with me and tell me–what I should expect, where I will be living, who my contacts are. The next year, I relay similar messages. I talk about the academic aspect of it. However, I haven't told anyone at the university about my experiences (of SGBV), so how can I tell the students going after me? I feel like crap about it. The person before me didn't tell me, and I didn't say anything. I said I had a great time, and it's not a lie. I just leave out a critical point.*

In the end, if a student has used strategies where she has successfully managed her situation and the people around her, what more can she do? At this point she may have kept herself "safe" from a physical assault or severe harassment. But her mitigation of SGBV is due to intense amounts of hidden labor at a high cost to her well-being.

> *After my supervisor heard of another incident of sexual harassment towards me, I tell her, "If you want to do something, deal with the people in your own institution." I tell her about the man from my practicum that has an affiliation with our university. "I've managed my experiences with him, so there isn't a specific incident to report. It's just a general vibe I get from him." When I come back to my university, I don't file a sexual harassment report (and neither*

*do the other women), because I don't have anything to report. What would I say, "Some guy gave me the creeps?" You can't report your feelings.*

Now that this student has mitigated many extreme forms of SGBV and "nothing" happened, she's left with no recourse. As she states, "*You can't report your feelings.*"

## Discussion

A phenomenology of global public health fieldwork aimed at understanding a woman graduate student's experience offers a starting place for universities to question their assumptions and current expectations related to student outcomes and well-being due to fieldwork. This phenomenological approach presents an opportunity to positively affect an individual's experience, and thus practice, allowing the student experience to act as a bridging mechanism–connecting, grounding, and amplifying their lived experiences. It builds shared knowledge and understanding through new meaning structures, language for an unfamiliar experience, or ways to describe, conceive of, and respond to this fieldwork. New meaning structures conceptualize women graduate students' lived experience when working to stay safe from SGBV through the structure of hidden labor. These structures include key phenomenological themes of *managing* and *laying traps* used to describe a student's extensive efforts to mitigate and avoid forms of SGBV. Through substantiating commonalities and calling out the pervasive nature of SGBV, we aim to create the opportunity for dialogue to ensure everyone has a safe space to learn and practice within the field of global public health. Our rigorous phenomenological research project aptly uses the embodied lived experience, adding valuable knowledge to global public health. Consequently, this knowledge offers an opportunity to create social change within academic institutions through structural and policy change rooted within students' lived experiences.

Throughout this story, a woman student's experience begins to surface ways that structural issues within the university can be problematic/problematized. The two distinct sets of influences create an intersection, different from a student studying in her home university or traveling abroad solo (Table 1). The precarity of being a graduate student combines with an unfamiliar environment to shift access to support networks and resources, creating a particular intersection with SGBV that has been inadequately addressed by universities sending students abroad for fieldwork. One consequence of feeling unable to share her story is that universities do not benefit from hearing these voices. Furthermore, academic institutions are not being held accountable as university structures and policies often compel women to remain silent or selectively share their story [34]. Thus, institutions are unable to act or respond to harm inflicted on their students. Furthermore, accountability is needed on both ends of the spectrum, from universities in the Global North as well as host institutions within LMICs. This necessitates open, productive, reciprocal, and equitable conversations and the ability for women to authentically share their experience as a critical first step in remedying the current situation.

The current lack of appropriate and available resources and supports has ethical and practical implications for students, the wider academic community, and the communities in which they work. Based on study findings and previous literature [4], universities and global health organizations need to restructure and reform the way in which institutional supports and resources are offered, delivered, and evaluated before, during, and after international student placements. This study offers a starting point for discussions aimed at implementing better protections and measures to keep all our colleagues safe, calling on universities to commit to providing more appropriate responses, offering greater support, and creating safer learning

environments. However, there is no easy checklist that a university can abide by. Rather, a comprehensive approach is needed, which includes protective policies and processes, as well as flexible procedures tailored to different student needs and circumstances. This type of hidden labor is hard to measure or quantify, as much of it is invisible, interwoven into structures and policies, and hidden through the strategies and techniques employed, requiring mental and emotional labor. Resources and supports may look wholly different from one student to another when responding to trauma and distress or helping students to best navigate challenging situations.

## Strengths and limitations

The character represented within this story evolved into many ways a student *can be*, only some of her stories capture her experience seeking safety as part of LGBT2Q+, racial, and multiple cultural communities and diverse personal background. However, many important identities are missing, and our attention is drawn to this silence, leaving us concerned to further suppression and absence of these ways of being-in-the-world. We recognize that safety and labor are compounded for those who experience systemic oppression due to their sexual orientation, gender identity, socio-economic status, abilities, racial and national identities, power relations, and their intersections [13,19]. There is ample scope for additional research necessary to identify safety needs and hidden labor for all students conducting global health fieldwork. This includes deeper inquiry focused on capturing the stories of students who identify as men, as well as the voices of all who have been historically oppressed, including those who are non cis gender, allowing untold stories of compounded hidden labor due to intersectionality into the literature. Research should further focus on university faculty and staff's experiences within this space, especially in relation to their supervision of students.

A criticism of phenomenology is that it is not attuned to power; moreover, the field of global health is rooted within a colonial history, with deeply embedded power structures that perpetuate and further exacerbate inequities [35,36]. While phenomenology is only one way to further understand global public health practice, this methodology offers the opportunity to identify experiences that have been silenced, suppressed, or taken for granted [37]. This methodological approach further recognizes and calls into question power dynamics within the research process by attempting to shift, deconstruct, and disrupt the researcher's epistemological authority [38]. However, as a co-creator within this phenomenological study, the first author recognizes that her positionality inherently impacted the way in which data from this study was created, analyzed, and interpreted, including how participants' stories have been shared. These concerns cannot easily be resolved through identifying or naming privileges or oppressions that one holds or endures; rather, it is something to continually reflect upon, allowing the researcher and the research to expose the shifts and transformations that need to occur. While phenomenology cannot capture any essence or phenomenon in totality, it is a complementary approach that does not aim to replace or refute other theoretical framings, but rather, to add further ways of knowing and understanding.

## Conclusion

While the experience of hidden labor due to SGBV applies to all graduate students in global health, this study captures a wealth of experience that clearly shows the burden on women is substantial, onerous, and remains largely unrecognized. Hidden labor is a woman graduate student's constant companion, both as she participates in global public health fieldwork and after her return. The remarkable extent of hidden labor revealed warrants further research on safety issues in global health research training and action by academic institutions. Most

notable are the sheer number of stories shared about women's hidden labor connected to SGBV, the continued silence and suppression of this experience, and the range and types of labor that cannot be described within a few simplified categories. Studies such as ours offer potential to better identify protections and supports needed to keep all our colleagues safe.

## Supporting information

**S1 Text. In-depth interview guide.**
(PDF)

## Acknowledgments

In gratitude for the support and mentorship offered through the Syme Training Fellowship in Work and Health, at the Institute for Work and Health.

## Author Contributions

**Conceptualization:** Corey McAuliffe, Erica Di Ruggiero.

**Data curation:** Corey McAuliffe.

**Formal analysis:** Corey McAuliffe.

**Investigation:** Corey McAuliffe.

**Methodology:** Corey McAuliffe, Ross Upshur, Daniel Sellen, Erica Di Ruggiero.

**Project administration:** Corey McAuliffe.

**Supervision:** Ross Upshur, Daniel Sellen, Erica Di Ruggiero.

**Validation:** Corey McAuliffe, Ross Upshur, Daniel Sellen, Erica Di Ruggiero.

**Writing – original draft:** Corey McAuliffe.

**Writing – review & editing:** Corey McAuliffe, Ross Upshur, Daniel Sellen, Erica Di Ruggiero.

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
