## [Decision Letter · Decision Letter 0]

29 Nov 2021

PGPH-D-21-00651

You can’t report your feelings: The hidden labor of managing threats to safety by women in global public health fieldwork

Dear Dr. McAuliffe,

Thank you for submitting your manuscript to PLOS Global Public Health. After careful consideration, we feel that it has merit but does not fully meet PLOS Global Public Health’s publication criteria as it currently stands. Therefore, we invite you to submit a revised version of the manuscript that addresses the points raised during the review process.

Please review and respond to the minor revisions suggested by your reviewers. 

We look forward to receiving your revised manuscript.

Kind regards,

Agomoni Ganguli-Mitra

Academic Editor

Journal Requirements:

1. Please include a copy of the interview guide used in the study, in both the original language and English, as Supporting Information, or include a citation if it has been published previously.

2. Please provide additional details regarding participant consent. In the ethics statement in the Methods and online submission information, please ensure that you have specified whether consent was informed.

3. In the online submission form, you indicated that "The data that support the findings of this study are available on request from the corresponding author, CM. The data are not publicly available due to their containing information that could compromise the privacy of research participants."

4. Please amend your detailed Financial Disclosure statement. This is published with the article, therefore should be completed in full sentences and contain the exact wording you wish to be published.

i) Please include all sources of funding (financial or material support) for your study. List the grants (with grant number) or organizations (with url) that supported your study, including funding received from your institution. 

ii). State the initials, alongside each funding source, of each author to receive each grant.

iii). State what role the funders took in the study. If the funders had no role in your study, please state: “The funders had no role in study design, data collection and analysis, decision to publish, or preparation of the manuscript.”

iv). If any authors received a salary from any of your funders, please state which authors and which funders.

Additional Editor Comments (if provided):

This is a very interesting and important paper. We would recommend that the comments from the reviewers are carefully attended to, which would place the paper in a very good position for acceptance.

Reviewers' comments:

Reviewer's Responses to Questions

**Comments to the Author**

1. Does this manuscript meet PLOS Global Public Health’s publication criteria? Is the manuscript technically sound, and do the data support the conclusions? The manuscript must describe methodologically and ethically rigorous research with conclusions that are appropriately drawn based on the data presented.

Reviewer #1: Yes

Reviewer #2: Yes

2. Has the statistical analysis been performed appropriately and rigorously?

Reviewer #1: N/A

Reviewer #2: N/A

3. Have the authors made all data underlying the findings in their manuscript fully available (please refer to the Data Availability Statement at the start of the manuscript PDF file)?

Reviewer #1: No

Reviewer #2: Yes

4. Is the manuscript presented in an intelligible fashion and written in standard English?

Reviewer #1: Yes

Reviewer #2: Yes

5. Review Comments to the Author

Reviewer #1: Dear colleagues

many thanks for such an interesting and unique paper. What would make it easier for readers to follow the stories is an added description of how the stories were developed in the methods section. Where the participants literally writing down aspects, or did the interviewers write the stories and then conduct member checking/have the participants read and make adjustments? It is not clear enough how the stories were really developed.

The limitations section should also include a discussion on what/how the researchers' identities may have played in the data collection and analysis, and if phenomenology was the best approach of choice in retrospect: what were benefits and risks by employing it?

Reviewer #2: This is a deeply refreshing and greatly enlightening piece of work that needs to be published and widely disseminated in the global health community. The authors have used an interesting and effective methodology to explore this topic and present the authentic and challenging experiences of participants navigating the field of global health to find a space for themselves. The title of this manuscript seems very appropriate and captures the essence of the emergent struggles of the participants. These experiences presented here are moving, and illuminate how rationalisation of such situations has become normalised in academic settings.

Overall, this is an important paper and I believe it should be accepted. I recommend a few suggestions below that might help to strengthen the manuscript for the readers.

1. A brief line providing some background context on the 2018 Workshop on Ethically Managing Global Health Fieldwork Risks.

2. A line or two describing how the participants were recruited and selected would be helpful.

3. Similarly, a brief description of the two methods of in-depth phenomenological interviews and guided writing exercises would help to make this approach more accessible to readers.

4. A critique of the larger landscape of global health field work in the global south within which these challenges of hidden labour are embedded would offer a deeper nuance to the discussion. It would be useful to also raise the aspect of accountability on both ends viz. universities in the Global North and host institutions in LMICs.

6. PLOS authors have the option to publish the peer review history of their article (what does this mean?). If published, this will include your full peer review and any attached files.

**Do you want your identity to be public for this peer review?** For information about this choice, including consent withdrawal, please see our Privacy Policy.

Reviewer #1: **Yes: **Dr. Melanie Boeckmann

Reviewer #2: No

---

## [Editor Report · Decision Letter 1]

21 Apr 2022

You can’t report your feelings: The hidden labor of managing threats to safety by women in global public health fieldwork

PGPH-D-21-00651R1

Dear Dr McAuliffe,

We are pleased to inform you that your manuscript 'You can’t report your feelings: The hidden labor of managing threats to safety by women in global public health fieldwork' has been provisionally accepted for publication in PLOS Global Public Health.

Best regards,

Agomoni Ganguli-Mitra

Academic Editor
